# Availability of Authorizations from EMA and FDA for Age-Appropriate Medicines Contained in the WHO Essential Medicines List for Children 2019

**DOI:** 10.3390/pharmaceutics12040316

**Published:** 2020-04-01

**Authors:** Jose-Manuel delMoral-Sanchez, Isabel Gonzalez-Alvarez, Marta Gonzalez-Alvarez, Andres Navarro-Ruiz, Marival Bermejo

**Affiliations:** 1Department of Pharmacokinetics and Pharmaceutical Technology, Miguel Hernandez University, 03550 San Juan de Alicante, Spain; jmoral@umh.es (J.-M.d.-S.); marta.gonzalez@umh.es (M.G.-A.); mbermejo@umh.es (M.B.); 2Institute of Molecular and Cellular Biology of Miguel Hernandez University, 03202 Elche, Spain; 3Pharmacy Service, General University Hospital of Elche, 03203 Elche, Spain; navarro_and@gva.es

**Keywords:** commercial availability, pediatrics, age appropriate, pharmaceutical preparations, safety, regulatory

## Abstract

Lack of age-appropriate commercially drug products availability is a common problem in pediatric therapeutics; this population needs improved and safer drug delivery. In addition, biopharmaceutic aspects, dosage requirements, and swallowing abilities demand pediatric forms different to adult formulations. The objective of this study was to evaluate the authorization availability from United States Food and Drug Administration (FDA) and European Medicines Agency (EMA) of oral essential medicines for children and analyze its age-appropriateness for oral administration in children. All oral drugs from 7th List of Essential Medicines for Children by World Health Organization (WHO) were selected. Availability of commercial drug products was collected from OrangeBook, Spanish drug product catalogue, British electronic Medicines Compendium, and the International Vademecum. Tablets, effervescent tablets, and capsules were considered as not age-appropriate forms. Liquid forms, powder for oral suspension, mini tablets, granules, and soluble films were considered as age-appropriate forms due to their flexibility. More than 80% of the studied drugs possess a commercial authorization in oral forms in both EMA and FDA. Nevertheless, around 50% of these formulations are not age-appropriate for most pediatric groups. This study shows the lack of age-appropriate medicines for children. More efforts are needed to improve development and approval of pediatric medicines.

## 1. Introduction

Nowadays, formulation research and development in the pediatric area remains essential and is required [1,2]. Governments allocate considerable efforts to promote the availability of age-appropriate, safe, and effective pediatric medicines by regulatory incentives [3,4]. Most drugs are often not appropriate for pediatrics due to the drug dosage form (tablets or capsules) and strengths. A lack of medicines specifically developed for children may be managed by preparing medicines extemporaneously or by manipulating dosage forms designed for adults, e.g., splitting tablets, crushing, and administering with food or liquid. Children cannot be considered small adults because of pharmacokinetic, pharmacodynamics, physiological, and anatomical differences.

Physiological aspects like the pH of the gastrointestinal tract [5,6] or expression of drug-metabolizing enzymes and transporters [7] are major facts for oral drug absorption and can alter the bioavailability of the administered drug [8]. These facts change with age and appear to be necessary to develop age-appropriate formulations. In respect with drug administration, children have different capabilities from adults, such as palatability or swallowing facility.

Marketed formulations are intended for adults predominantly, so pharmaceutical compounding and manipulations are often necessary, such as crushing tablets, dispersing content capsule or crushed tablets in liquid excipients, or using parenteral formulation for oral administration [9,10].

In connection with appropriate drug dosage forms for oral administration, the most suitable forms are syrups, oral solutions, or suspensions. They ensure the administration of different doses with a single medicine and the possibility of mixing them with food or beverages [11,12].

Nowadays, new solid oral formulations are emerging [13]. These novel pediatric specific formulations include: Mini tablets, oral-soluble films, oral powders and granulates, orally disintegrating tablets, chewable tablets, scored tablets, and sprinkle capsules [14]. New solid oral forms could be age appropriate if the flexibility of dosing can be achieved and may avoid the use of harmful excipients and palatability issues for liquid medicines.

Apart from the lack of age-appropriate medicines, there is almost a total absence of medicines approved for neonates and common formulations are used because there are no suitable alternatives [4].

World Health Organization (WHO) Essential Medicines List for children (EMLc) considers the efficacious, cost-effective, and safest drugs for priority pediatric pathologies for a basic healthcare system [15]. Listed drugs with immediate release oral forms from 7th Edition (2019) of EMLc were selected to examine the authorization availability of age-appropriate medicines from the European Medicines Agency (EMA) and United States Food and Drug Administration (FDA).

## 2. Materials and Methods

All drugs with an available oral administration route from 7th List of Essential Medicines for Children [15] were selected.

Drug products’ information from FDA was collected in July of 2019 from OrangeBook [16]. Information from EMA was listed from Spanish drug product catalogue (CIMA) [17], British electronic Medicines Compendium (eMC) [18], and the International Vademecum [19] due to these databases being public access formularies. International Vademecum was selected because it is a database with data from the member countries of EMA.

Selected drugs were classified into orally available if the drug is commercially available in any drug dosage form of oral administration.

Solid dosage forms like tablets, effervescent tablets, or capsules were considered as not age-appropriate forms for pediatrics due to their poor versatility for dosage administration. In the present study, pediatric age range was considered as 0 to 12 years because it is the contemplated age range in EMLc.

Drug dosage forms like syrups, oral solutions or suspensions, chewable tablets for pediatrics, granules, oral-soluble films, minitablets, or powder for oral suspension were assigned to age-appropriate forms for pediatric population. Furthermore, the first authorization year in FDA and EMA of age-appropriate forms was included.

Additionally, selected drugs were classified into WHO Anatomical Therapeutic Chemical (ATC) classification system (first level—anatomical main group), the recommended system by WHO for international drug utilization studies [20,21], and an intragroup analysis was done about drugs with age-appropriate forms.

The percentages of several analyses were the result of the following equations:% availability = (orally commercial drugs available)/(selected drugs),(1)
% age-appropriate = (drugs with oral pediatric age-appropriate form)/(orally commercial drugs available),(2)
% age-appropriate (per therapeutic group) = (drugs with oral pediatric age-appropriate form from therapeutic group x)/(drugs within therapeutic group x).(3)

## 3. Results

### 3.1. Availability and Suitability of Oral Formulations

Table A1 shows all 149 drugs included as essential medicines in children for oral administration and its classification as commercially available or age-appropriate oral formulation for pediatrics in EMA or FDA. Additionally, new solid oral forms and age-appropriate forms that emerged in order to meet requirements years after their first authorization are highlighted.

Figure 1 shows the authorization availability of orally commercial forms both in EMA and FDA. Figure 2 illustrates the age appropriateness of the orally commercial formulations both in EMA and FDA.

### 3.2. Therapeutic Class Distribution

Figure 3 shows the therapeutic class distribution by the ATC Classification System regarding the analysis per groups, Therapeutic classes to which the selected oral drugs belong were: A: Alimentary tract and metabolism; B: Blood and blood forming organs; C: Cardiovascular system; H: Systemic Hormonal preparations; J: Anti-infective for systemic use; L: Antineoplastic and immunomodulation agents; M: Muscular-skeletal system; N: Nervous system; P: Antiparasitic products; R: Respiratory system; and V: Various.

Figure 4 shows the percentage of the age appropriateness of available drugs with respect to ATC classes.

## 4. Discussion

Specific development of drug products for pediatrics has been inappropriate because of the lower prevalence of diseases in children in comparison with adults, and also relates to commercial reward. The scarcity of resources with regard to pediatric pharmacotherapy is an acknowledged gap by worldwide governments and regulatory agencies [22]. The first step was taken in 1970s when FDA stated that most prescription drugs were administered empirically and it called for innovative programs to provide pediatric information [23] and recognized that excluding children from clinical trials was an unethical fact and could create risk situations [4]. Since then, many useful efforts have been devoted to promoting research in pediatric pharmacotherapy. Even now, more than 40 years later, age-appropriate medicines for children are still not available. In the present study, only drugs from the List of Essential Medicines for Children of WHO were analyzed, but there are considerably more drugs to analyze that are also not available and are used in off-label conditions.

The commercial availability of oral forms of selected drugs proved to be high (84.6% in EMA vs. 79.9% in FDA) (Figure 1).

Differences in the formularies or catalogues of all countries that comprise EMA were an identified limitation. The presence of a drug in one of the drug catalogues does not mean that the product is in fact available in all EMA countries. An additional limitation was not being able to access all the official catalogues of the member countries.

Another limitation was that the essential list by WHO is made for worldwide countries, so all drugs listed are perhaps not relevant for all clinical settings. For example, fexinidazole is an antiparasitic drug indicated for African Chagas without authorization available in the FDA nor in EMA (Table A1). There are several factors that determine the availability of therapies in a region or market, including disease prevalence and an available patient population to complete development.

Highlighted drug forms (Table A1), such chewable tablets (i.e., Ibuprofen or Lamotrigine), oral-soluble films (i.e., Ondansetron), or prolonged release granules (i.e., Valproic acid), were found as authorized medicines with age-appropriate and innovative forms. Other updated formulations of drugs, such as Propranolol (oral solution 4mg/mL) and Methotrexate (oral solution 2 mg/mL), were found and they were developed to meet special pediatric requirements and pathologies (propranolol for infantile hemangioma and Methotrexate for acute lymphoblastic leukemia and polyarticular juvenile idiopathic arthritis). It should be noted that these new formulations were developed and authorized from 2007, when the EMA legislation took effect and the FDA was more effective, with the formation of the Pediatric Review Committee.

In regard to the ATC class distribution (Figure 3), anti-infective drugs of J class (anti-infective for systemic use) P class (antiparasitic products) constitute the largest group (more than 50%).

It is noteworthy the an extremely lower percentage of age appropriateness of antiparasitic formulations is one of the majority groups (Figure 4). Groups B (blood), P (antiparasitic products), and R (respiratory system) are the ones in the FDA area showing a lack of age-appropriate formulations. Conversely, N class (nervous system) drugs had the highest percentage of age-appropriate forms in both EMA and FDA, followed by J (anti-infective for systemic use) and M (muscular-skeletal system) groups.

Furthermore, the presence of drugs used in the treatment of neglected diseases (21.7% of included drugs) should be noted. While in most cases these are commercially available, very few are age-appropriate dosage forms for pediatric oral administration, as well as posological adjustment. Frequently, formulations are compounded from adult drug products to avoid the problem, but as explained above, this practice is not considered advisable but unavoidable, and can lead to biopharmaceutical and safety problems [12,24].

Despite the high accessibility of oral drug products, a gap can be identified regarding the suitability of drug dosage forms to pediatrics (52.3% in EMA vs. 45.6% in FDA) (Figure 2). Pediatric populations need maximal dosing flexibility, palatability, and safety [25,26].

Regulatory authorities are aware of the lack of age-appropriate forms and the need to use extemporaneous formulations from active product ingredients or even from manipulation of adult dosage forms as it was recognized in the Reflection Paper: Formulations of choice for the pediatric population by EMEA in 2006 [12]. A potential risk for the consistent performance of those compounded formulations is the effect of excipients on the release, transit, and absorption of low solubility and/or low permeability drugs. The selection of suitable excipients and its age-related safety profiles are especially critical in drug product development and pharmaceutical compounding intended for neonates and young children [27].

Another potential risk, associated with compounding, is the solution osmolarity in liquid formulations, which eventually may affect the membrane permeation rate [28]. Any change on absorption could be in particular problematic in drugs with a narrow therapeutic range [25]. The recognition nowadays that excipients could not be “inert” components but have an effect of gastrointestinal motility, permeability, or fluid balance [26,29] is of especial relevance in children due to its rapid developmental changes in intestine physiology and because our knowledge at this level is still scarce [29].

The Biopharmaceutics Classification System (BCS) is a widely evolved and used tool in the development of medicines in adults [30] as well as physiologically based pharmacokinetics (PBPK) modelling and the dissolution test (in vitro) [31]. The development of a pediatric biopharmaceutics classification system (pBCS) could help to identify those drugs for which harmonization of compounded formulas would be advisable [6,28,29] as their absorption rate/extent is particularly sensitive to the effect of excipients on drug solubility, permeability, or the dissolution rate.

## 5. Conclusions

This quantitative evaluation confirms the need for improvements in drug delivery in pediatrics and the lack of age-appropriate medicines in many therapeutic areas. Currently, real efforts are being made to improve the development and approval of drug products aimed for children because of global requirements. These formulations must be able to adapt to pediatric oral biopharmaceutics and capabilities.

Although it is a difficult task to carry out, this paper calls for suitable pediatric formulations that can be orally administered in an appropriate form based on dose flexibility, swallowability, and palatability. In addition, thinking regarding older products and an attempt to develop age-appropriate medicines, as with the drugs that have already been achieved by new oral solid formulations, should be prioritized.

## Figures and Tables

**Figure 1 pharmaceutics-12-00316-f001:**
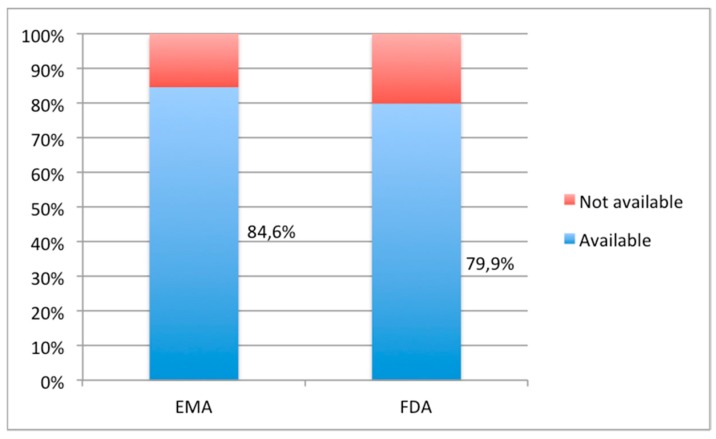
Percentage of oral dosage forms in Essential Medicines List for children (EMLc) 2019 listed as authorized medicines by European Medicines Agency (EMA) and United States Food and Drug Administration (FDA).

**Figure 2 pharmaceutics-12-00316-f002:**
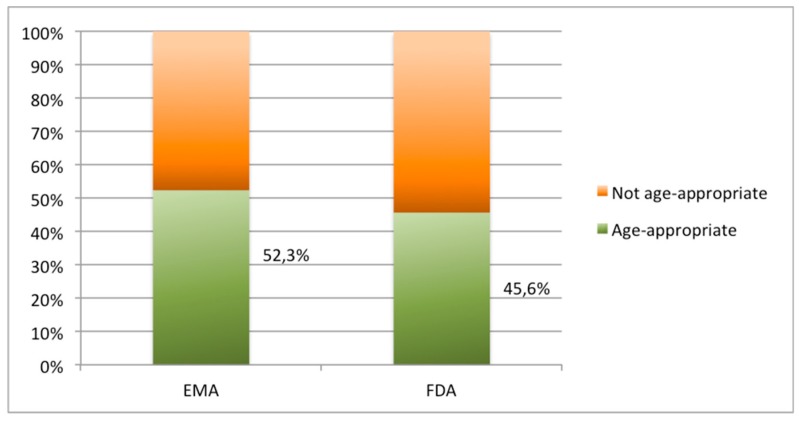
Percentage of oral dosage forms in EMLc 2019 listed as authorized age-appropriate medicines by EMA and FDA.

**Figure 3 pharmaceutics-12-00316-f003:**
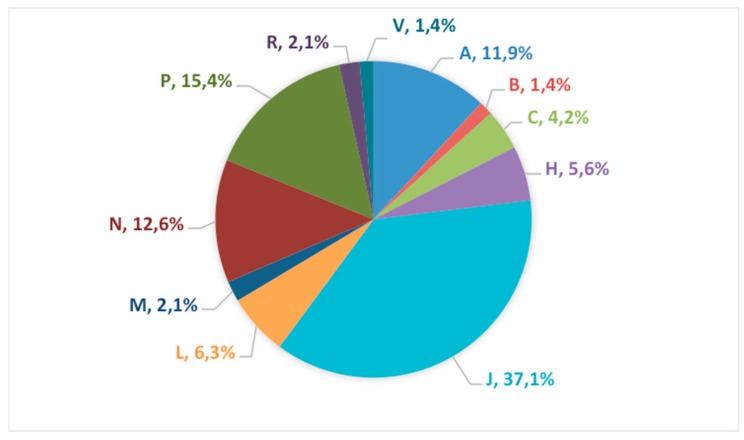
Percentage of oral drugs listed in EMLc by Anatomical Therapeutic Chemical (ATC).

**Figure 4 pharmaceutics-12-00316-f004:**
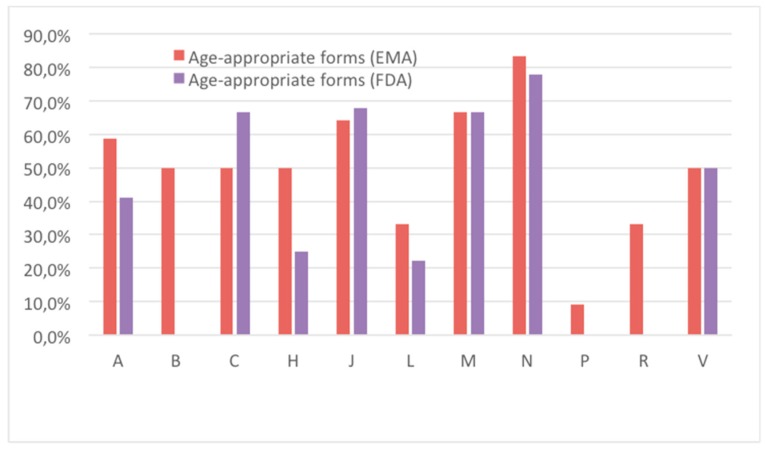
Comparison of the availability of authorized oral age-appropriate dosage forms within therapeutic (ATC) classes.

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
