# Peer review of "Availability of Authorizations from EMA and FDA for Age-Appropriate Medicines Contained in the WHO Essential Medicines List for Children 2019"

_pharmaceutics, 2020, doi:10.3390/pharmaceutics12040316_

Round 1
Reviewer 1 Report
General: This study is a nice review of pediatric products and the WHO Essential Medicines List. However, the authors have neglected to put this work into perspective in relation to the legislation in the US and in Europe. That analysis will actually help to focus the results and conclusion.
Specific comments:
Throughout: Some improvement of the English is needed, so this should be revised by someone familiar with scientific writing.
Methods: The authors should incorporate a timeline for each of the products related to: the year(s) of pediatric legislation in the US and Europe, the year the product was approved for adult use and pediatric use (if approved for pediatric use); and any outstanding post marketing requirements for the product (pediatric products may be in development).
Conclusion: My suspicion is that a reanalysis using my suggestion above will focus the concern on the products approved before 2007, when the EMA legislation took effect and the US legislation was more effective (formation of the FDA's Pediatric Review Committee). The authors can then focus their thinking about how to deal with those older products, which is difficult.
Author Response
Thanks for your suggestions that I am glad to say that they have improved tha article.
Suggestions have been added in the doc, tables, discusion and conclusions:
- The year of authorization in FDA and in EMA (the year of the first country that authorized it from the analyzed ones) of the age-appropriate forms was added
- The table shows the innovative formulations (films, orodispersibles, granules, etc.) or formulas that have been developed many years after the first authorization to cover specific pediatric needs. New results of authorized products after 2007 as a cut-off point have been added to the discussion.
- The discussion has been completed with the reviewer's suggestions
"Highlighted drug forms (Table A.1) such chewable tablets (i.e. Ibuprofen or Lamotrigine), oral soluble films (i.e. Ondansetron), or prolonged release granules (i.e. Valproic acid) were found as authorized medicines with age-appropriate and innovative forms. Another updated formulations of drugs as Propranolol (oral solution 4mg/mL) and Methotrexate (oral solution 2mg/mL) were found and they were developed to meet special pediatric requirements and pathologies (propranolol for infantile hemangioma and, Methotrexate for acute lymphoblastic leukemia and polyarticular juvenile idiopathic arthritis). Should be noted that these new formulations were developed and authorized from 2007, when the EMA legislation took effect and the FDA was more effective with the formation of the Pediatric Review Committee."

Reviewer 2 Report
See attachment for comments.

Author Response
All proposed ideas and comments from Reviewer 2 have beed add to the manuscript.
see final doc

Round 2
Reviewer 1 Report
My previous comment was to suggest that the analysis be redone to assess pre-2007 versus post-2007. I believe that this may change your assessment and overall conclusion.
Author Response
The authorization date of a new formulation can be obtained because it is not confidential information. However, the authorization of use in pediatrics is not easily available.
We improved so much the article based on the previous contribution of the Reviewer1 because we added information about data of authorization of age-appropriate medicines. Thanks for this suggestion because with this modification, it can be seen that the innovative formulations emerged after 2007 when all regulatory agencies focus on this purpose: EU Paediatric Drug Regulation was enacted and the PIPs were established and in FDA, the Paediatric Review Committee was constituted. As we discussed in our previous modification, formulations appear to adapt to the capabilities of children and others for specific paediatric pathologies:
“Highlighted drug forms (Table A.1) such chewable tablets (i.e. Ibuprofen or Lamotrigine), oral soluble films (i.e. Ondansetron), or prolonged release granules (i.e. Valproic acid) were found as authorized medicines with age-appropriate and innovative forms. Another updated formulations of drugs as Propranolol (oral solution 4mg/mL) and Methotrexate (oral solution 2mg/mL) were found and they were developed to meet special paediatric requirements and pathologies (propranolol for infantile haemangioma and, Methotrexate for acute lymphoblastic leukaemia and polyarticular juvenile idiopathic arthritis). Should be noted that these new formulations were developed and authorized from 2007, when the EMA legislation took effect and the FDA was more effective with the formation of the Paediatric Review Committee.”
We are aware of an article in which the dates of authorization of paediatric were evaluated and the influence of the European paediatric regulation was analysed:
The influence of the European paediatric regulation on marketing authorisation of orphan drugs for children. Kreeftmeijer-Vegter et al. Orphanet Journal of Rare Diseases 2014, 9:120
In that paper the authorization dates for paediatrics are obtained because the EMA provides these data to the authors as they said literally: “EMA kindly provided us with a list of all ODDs from 2000 until December 2012 with designation date and numbers…”. They concluded: "The
EU Pediatric Drug Regulation did not increase the number of orphan drug
designations with potential pediatric indications nor did it lead to more
marketing authorizations for pediatric indications." But, that
paper was published in 2014
Only 5 drugs studied in that paper has been included in our article (149 orals included in the 7th List of Essential Medicines for Children):
- Caffeine citrate: july 2009
- Hydroxycarbamide: june 2007
- Imatinib: november 2011
- Mercaptopurine: march 2013
- Zinc: april 2004
As we have mentioned before, we do not have access to the EMA and / or FDA
pediatric authorization data, but if reviewer can help us about how to get them, we will be glad to analyse and add this analysis from both regulatory agencies.
Round 3
Reviewer 1 Report
The authors have addressed the reviewers' comments.